# Automatic Detection of Aerobic Threshold through Recurrence Quantification Analysis of Heart Rate Time Series

**DOI:** 10.3390/ijerph20031998

**Published:** 2023-01-21

**Authors:** Giovanna Zimatore, Cassandra Serantoni, Maria Chiara Gallotta, Laura Guidetti, Giuseppe Maulucci, Marco De Spirito

**Affiliations:** 1Department of Theoretical and Applied Sciences, eCampus University, 22060 Novedrate, Italy; 2CNR Institute for Microelectronics and Microsystems (IMM), 40129 Bologna, Italy; 3Neuroscience Department, Biophysics Section, Università Cattolica del Sacro Cuore, 00168 Rome, Italy; 4Fondazione Policlinico Universitario A. Gemelli IRCCS, 00168 Rome, Italy; 5Department of Physiology and Pharmacology “Vittorio Erspamer”, Sapienza University of Rome, 00185 Rome, Italy; 6Department Unicusano, Niccolò Cusano University, 00166 Rome, Italy

**Keywords:** aerobic transition, determinism, heart rate, non-linear time series analysis, physical exercise, ventilatory threshold

## Abstract

During exercise with increasing intensity, the human body transforms energy with mechanisms dependent upon actual requirements. Three phases of the body’s energy utilization are recognized, characterized by different metabolic processes, and separated by two threshold points, called aerobic (AerT) and anaerobic threshold (AnT). These thresholds occur at determined values of exercise intensity(workload) and can change among individuals. They are considered indicators of exercise capacities and are useful in the personalization of physical activity plans. They are usually detected by ventilatory or metabolic variables and require expensive equipment and invasive measurements. Recently, particular attention has focused on AerT, which is a parameter especially useful in the overweight and obese population to determine the best amount of exercise intensity for weight loss and increasing physical fitness. The aim of study is to propose a new procedure to automatically identify AerT using the analysis of recurrences (RQA) relying only on Heart rate time series, acquired from a cohort of young athletes during a sub-maximal incremental exercise test (Cardiopulmonary Exercise Test, CPET) on a cycle ergometer. We found that the minima of determinism, an RQA feature calculated from the Recurrence Quantification by Epochs (RQE) approach, identify the time points where generic metabolic transitions occur. Among these transitions, a criterion based on the maximum convexity of the determinism minima allows to detect the first metabolic threshold. The ordinary least products regression analysis shows that values of the oxygen consumption VO_2_, heart rate (HR), and Workload correspondent to the AerT estimated by RQA are strongly correlated with the one estimated by CPET (r > 0.64). Mean percentage differences are <2% for both HR and VO_2_ and <11% for Workload. The Technical Error for HR at AerT is <8%; intraclass correlation coefficients values are moderate (≥0.66) for all variables at AerT. This system thus represents a useful method to detect AerT relying only on heart rate time series, and once validated for different activities, in future, can be easily implemented in applications acquiring data from portable heart rate monitors.

## 1. Introduction

Heart rate (HR) time series are widely used to characterize physiological states, fitness level, and athletic performance [1]. For example, features like HR peak and maximal oxygen uptake (VO_2max_) estimated from HR time series are used as indexes of Cardiorespiratory Fitness [2,3].

Personalization of exercise intensity is based on the submaximal threshold concept [4]. The term threshold refers to a transition in the metabolic process during incremental intensity exercises [5]. The aerobic threshold (AerT) [6] indicates the change from aerobic metabolism (production of energy using oxygen) to anaerobic metabolism (production of energy without oxygen) occurring at particular exercise intensity values. On the other side, the transition in which a ventilation increase takes place, because of increasing carbon dioxide production, is known as the anaerobic threshold (AnT) [5,6]. AerT detection is widely used not only in sports, but also in medical settings, to study sedentary people and active recovery processes. Instead, AnT detection mostly concerns amateur and professional athletes to develop customized training plans [3,4]. The gas exchange testing (cardiopulmonary exercise test, CPET) and the blood lactate sampling are the standard techniques for threshold detection, occurring by assessing cardio-respiratory performance at rest and during exercise [7], or evaluating blood lactate concentration [8]. Although these methods (the gas exchange testing and the blood lactate sampling) are efficient, their widespread use in real life settings is not feasible, being impractical (as they require a laboratory equipped with specific devices and trained operators) or highly invasive (as they require blood draws) [2].

An automated determination of the metabolic thresholds by using HR series is therefore highly desirable. Nowadays, wearable smart devices allow the continuous daytime measurement of HR in the most diverse contexts. Detection of thresholds is in principle possible by using non-linear time series analysis, recently reviewed in [2]. Among the most promising studies, a recurrence quantification analysis (RQA)-based approach was used to identify, in adult subjects, both aerobic and anaerobic thresholds during an incremental test [9]. We used RQA on heart rate time series to estimate the AerT in obese subjects [10] and, successively, in healthy young subjects. These methods allowed for the determination of both aerobic and anaerobic thresholds. However, though relying only on HR time series, they require visual inspection and analyses by experts for threshold identification. To promptly measure these quantities and track their time variations without the need to rely on expensive equipment and specialists, relying only on portable heart rate monitors, here we present a method to automatically identify the aerobic threshold (AerT) by an RQA-based approach. This approach relies on the HR time series, acquired during a sub-maximal incremental exercise test (Cardiopulmonary Exercise Test, CPET) on a cycle ergometer [11].

Since performing the RQA feature evidenced by Recurrence Quantification by Epochs (RQE), we found out that the minima of determinism identify the time points where generic metabolic transitions occur. The hypothesis is that among these transitions, a criterion based on the maximum convexity of the determinism minima allows to detect the first metabolic threshold in a fully automatic way.

We tested the method on HR time series, acquired from a cohort of competitive and recreational athletes during a sub-maximal incremental exercise on a cycle ergometer.

## 2. Materials and Methods

### 2.1. Data Set

Six competitive rowers (group A, 15.33±2.16 ys), 8 recreational rowers (group B, 16.0±2.0 ys), and 13 athletes that practice other recreational sports (group C, 14.8±1.6 ys) were involved in the study. Competitive rowers of group A were athletes who completed at least 5 workouts per week and had participated in regional and/or national competitions in the preceding year. Recreational rowers of group B went to rowing activities twice a week. The group C consisted of individuals who engaged in recreational activities other than rowing (twice a week).

Clinical checks were performed on all individuals to rule out any physical activity-related adverse effects. However, a medical certificate was required for competitive and noncompetitive sports activities. Neuropathy, autonomic dysfunction, and cardiovascular disorders were considered as exclusion criteria. Before the study began, all volunteers (or their parents if they were under the age of 18) signed a written informed consent form. The CAR-IRB—University of Rome “Foro Italico” Committee approved this study (Approval N° CAR 37/2020) in conformity with the Declaration of Helsinki.

The number of points of time series (Npoints), collected during the incremental exercise, can be different.

### 2.2. Incremental Exercise on Cycle Ergometer

The following protocol was used for the incremental exercise test.

(i)The test occurred in the morning between 9:00 and 12:00 a.m. in identical conditions (21–22 °C; humidity 50–60%). At least 90 min before the test, the subjects ate their usual breakfast. Before the exercise, participants were subjected to clinical and anthropometric evaluations. Then, they had to maintain a 60–70 revolutions per minute (rpm).(ii)Participants were seated on the bike for 1 min and then cycled unloaded for 1 min (0 W). The workload was then increased for the A group by 20 W/min (protocol 1) and for the B and C groups by 15 W/min (protocol 2). These different protocols were carried out to keep a similar amount of exercise time [9,12]. All participants carried out the test under observation of the staff, for the correct execution of a graded physical exercise on the cycle ergometer. The operator kept track of any anomalies in execution.(iii)During the test, participants’ perceptions of physical effort were measured 15 s before the workload increase using the OMNI Scale of Perceived Exertion (0–10 scale) [13]. The participants were asked to assess their perceived level of effort on a scale from 0 (very easy) to 10 (extremely difficult). The test stopped when one of the following requirements was met: a score of 10 on the OMNI Scale of Perceived Exertion, a respiratory exchange ratio of 1.1, or 90% of the subject’s estimated HRmax (beats/min). An automated gas analyzer (Quark RMR-CPET Cosmed^TM^, Rome, Italy) [11] monitored oxygen consumption (VO_2_, mL/min), carbon dioxide generation (VCO_2_, mL/min), and pulmonary ventilation (VE, mL/min) while the HR was recorded breath-by-breath by a chest belt (HRM-Dual TM, Garmin^®^, Olathe, KS, USA) [9]; for details see Section 2.3.1.

HR time series were recorded breath-by-breath by CPET. Since the time intervals between two consecutive points in the recorded breath-by-breath time series were not equal, HR was resampled every 2 s (the most frequent time interval).

### 2.3. Detection of the Individual Ventilatory Threshold (AerT)

In Figure 1a, the representative individual graph of HR and the corresponding power of an agonist subject for the entire exercise duration is reported. The HR (Figure 1a) time series is analyzed by a script realized in Python.

In the following the steps of the algorithms are reported:(i)HR time series intervals, corresponding to the initial and final workload increment are ruled out from the analysis since the RPM value is not constant and out of the inclusion interval 60–70 rpm (Figure 1b).(ii)RQA ([9,10]) is applied to analyze acquired time series data. RQA epoch-by-epoch analysis (RQE) is performed to find determinism percentage (*DET*) of HR time series, which is the percentage of recurrence points which form diagonal lines in the recurrence plot. RQA epoch-by-epoch analysis is performed on intervals of width Δw = 100 points (200 s) with the following input parameters: embedding = 7, shift = 1, radius = 5, line = 4 (for more details see [14]).(iii)the AerT is found by selecting the most convex minimum of *DET*:
$$DET_{\min}=\max\left(\frac{d^{2}DET\left(t\right)}{d^{2}t}\right)$$
among all relative minima of *DET* (Figure 2b, colored points), called *DET*_min_ retrieved at the time point t_VT1_. Only minima with a value *f”(t) >mean(f”(t)) +2 × SD(f”(t))* (named Cut-off) are detected and counted. HR, Workload, and VO_2_ values correspondent to t_VT1_ are named W_RQA, HR_RQA, and VO_2__RQA, respectively.

#### 2.3.1. Gas Exchange Method (GEx Method)

In this paper, the gold standard for detecting the AerT threshold was based on gas exchange analysis (GEx method) revealed by a CPET device (Cosmed^®^) [11].

The ventilatory equivalent of oxygen (VE/VO_2_) was plotted as a function of VO_2_ to determine the point where the VE/VO_2_ showed its lowest value during the incremental exercise test. This allowed us to estimate the individual ventilatory threshold for each subject [15,16]. The AerT was defined by the level of VO_2_ at the lowest value of the ratio VE/VO_2_ [10].

To assess the value of HR and VO_2_ corresponding at the threshold, their mean values on the last 30 s of the workload were reckoned, respectively. The Workload, HR, and VO_2_ values obtained by GEx-methods are those reported in [9] (named W_GEx, HR_GEx, and VO2_GEx, respectively).

### 2.4. Statistical Analysis

A Kolmogorov–Smirnov Test of Normality was performed to confirm that VO_2_, HR, and Workload values are normally distributed (specifically: VO_2_RQA_ (D = 0.165, *p* = 0.48), HR__RQA_ (D = 0.845, *p* = 0.98), and Workload__RQA_ (D = 0.158, *p* = 0.46).

To assess main effects of methods (RQA vs. GEx) and methods-by-groups (A, B, and C) interaction effects for HR, VO_2_, and Workload at AerT, the repeated measures analysis of variance (RM ANOVA) was used. The Ordinary Least Products (OLP) regression analysis [17] was used to assess the agreements between RQA and GEx methods for HR, VO_2_, and Workload at AerT.

To identify fixed and proportional biases, the OLP regression equation’s coefficients of determination (R^2^) (that is the square of correlation r) and regression parameters (slope and intercept) with 95% confidence intervals (CI) were determined.

The following criteria were adopted to interpret the magnitude of the correlation r between the estimates: <0.1 trivial, 0.1–0.3 small, 0.3–0.5 moderate, 0.5–0.7 large, 0.7–0.9 very large, and 0.9–1.0 almost perfect [18].

When the 95% CI contained the value 1 for the slope and the 0 for the intercept, the hypothesis of proportional and fixed bias was rejected. For HR, VO_2_, and Workload, the percentage differences between the RQA and GEx methods at AerT (reported as mean and range values) were calculated. The validity of the RQA-based method was also evaluated by comparing HR, VO_2_, and Workload at AerT between the RQA-based vs. GEx method with a paired samples *t*-test. Typical percentage Error (TE) was calculated by dividing the standard deviation of the difference percentage by √2. As a parameter for criterion validity of the RQA method compared to the GEx method at AerT for all variables (HR, VO_2_, and Workload), the intraclass correlation coefficients (ICC) [19] was used.

Furthermore, the magnitude of the differences between RQA-based and GEx methods was assessed using effect size statistics (as Cohen’s d) with 90% confidence interval (CI) and percentage change [19]. If d < 0.2, the effect size was classified as trivial, if d falls in the range 0.2–0.6 as small, in the range 0.6–1.2 as moderate, in the range 1.2–2.0 as large, finally if d > 2.0 was classified as very large [20].

ANOVA 1-way was conducted on the number of exercise increment (Nstep), on the number of minima (Nmin), and the number of time series’ points (Npoints) to verify if there is any difference on these measures among groups.

Statistical significance was defined as *p* ≤ 0.05. All statistical analysis was performed by SPSS version 24.0 software (SPSS Inc., Chicago, IL, USA).

## 3. Results

### 3.1. Determination of AerT through RQA Approach

As a first step, the determinism percentage (DET%) epoch-by-epoch was obtained for each individual HR time series. In Figure 2a, the percent of determinism (DET(t)) and second temporal derivative ($\frac{d^{2}DET\left(t\right)}{d^{2}t}$) are reported. The second derivative allows to select those with higher convexity (*DET*_min_). As reported in Section 2.3, the aim of this work is to show that the most convex minimum is associated to AerT (t_VT1_). This minimum is indicated by a red dot in Figure 2b.

In Figure 2b are reported the determinism percentage (black points) and the workload (light blue line). The red vertical line corresponds to AerT (t_VT1_); the concurrent behavior of *DET* epoch-by-epoch is used to study the increase in correlation [10].

### 3.2. Detection of Minima Corresponding to the First Ventilatory Threshold

We automatically detected AerT on the cycle ergometer during an incremental exercise test and the agreement between the GEx method and the automatic procedure was evaluated for the values of Workload, HR, and VO_2_ at AerT.

Results are presented in Table 1. OLP regression analysis showed that the values of HR and Workload at AerT for GEx and the RQA method have strong correlations (r > 0.64). Mean percentage differences between the two evaluation methods are <2% for both HR andVO_2_ and <11% for Workload, and their difference are not statistically significant (*p* > 0.05) at AerT. The Technical Error (TE) for HR is <8%, for VO_2_, it is around 16%, and for Workload, it is around 15%. These values, even if higher than HR, are acceptable values if performed as automatic detection [18,21,22]. The intraclass correlation coefficients (ICC) values were moderate (≥0.66) for all variables at AerT [19]. The effect size d (Cohen) was trivial for HR and VO_2_ (Table 2).

The OLP regression analysis plots of Workload, HR, and VO_2_ values at AerT for both methods (GEx and RQA) are graphically shown in Figure 3a–c, respectively. In each graph, the OLP line of best fit is represented with a solid line, while the dotted line represents the line of unity. We also reported the equation, the correlation coefficient (r), and the absolute mean differences (Table 2).

It is worthwhile to note that the sample is very general since we are considering three groups with very different fitness levels.

### 3.3. Groups Comparison

The twenty-seven subjects considered in this work belong to three different groups, A (Competitive rowers), B (recreational rowers), and C (subjects that practice other sports), and have different fitness levels and a different value of the maximum heart rate.

No significant differences were reported for all other anthropometric variables among the three groups (see Table 3).

Firstly, the workload increments were counted, and this number (Nstep, see blue stairs line in Figure 1), reported in Table 4, was not significantly different among groups (*p* > 0.05).

Successively, by considering the more convex minima (over the dashed red cut-off line in Figure 2), we observed that the number of these minima was not significantly different among groups (*p* > 0.05).

Finally, it was verified that there is no significant difference in the number of points (Npoints, see black points in Figure 1) among the three groups (Table 4).

To check whether the method is biased by the fitness level, two factors (Gex vs. RQA) repeated measured ANOVA for three groups (A, B, and C) was conducted on Workload, HR, and VO_2_ at AerT. As reported in Table 5, there is no significant differences between the quantities calculated with the two methods, and no methods-by-groups interaction effects were detected as reported in Table 5.

## 4. Discussion

In the last ten years, specific attention was paid to non-linear methods to analyze biological complex systems. Different non-linear methods (Poincarè, RQA, DFA) were used to detect physiological transitions ([2,22]), with the final aim of the real time monitoring of physiological observables by using wearable devices [23]. In this context, we developed a new automatic procedural method for detecting aerobic thresholds based on RQA analysis of HR and confirmed the validity of the RQA method in threshold detection when compared to the GE method on a dataset of 27 people with different fitness levels.

This determination is fundamental in athletes to monitor their health being correlated to fat oxidation [24]. We found that the minima of determinism, an RQA feature calculated from the Recurrence Quantification by Epochs (RQE) approach, identify the time points where generic metabolic transitions occur. Among these transitions, a criterion based on the maximum convexity of the determinism minima allows to detect AerT with a low TE. The ordinary least products regression analysis shows that values of the oxygen consumption VO_2_, heart rate (HR), and Workload corresponding to the AerT estimated by RQA are strongly correlated with the one estimated by CPET (r > 0.64). Mean percentage differences are <2% for both HR and VO_2_ and <11% for Workload. The Technical Error for HR at AerT is <8%; intraclass correlation coefficients values are moderate (≥0.66) for all variables at AerT. Differences between RQA and GE methods assessed using effect size statistics (as Cohen’s d) were trivial (<0.2) for all parameters. We moreover found that this approach is unaffected by the individuals’ degree of physical fitness and came from the absence of methods-by-group interactions in all variables at AerT.

With respect to the previous studies reported by our group on the same topic [9,10], the analysis method underwent a net improvement, mainly for two aspects: (i) the automation of the method allowing the real time determination of AerT without the need of visual inspection, and (ii) the absence of the pre-processing: no detrend was carried in the automatic detection proposed in order to delete a possible source of dependence from input parameter selection. Both these improvements favor the implementation of this method on portable cardiac monitors. We have to point out that this implementation is not straightforward. In this work, the use of a cycle-ergometer allows for an accurate workload control in terms of intensity and duration. In these ideal conditions, it is possible to define parameters needed to calculate the RQA plot as already reported in previous work [9,10]. Therefore, we have to further test if the parameter choice can be affected by the type of exercise protocol.

At present, we can say that since the groups underwent a different workload increase (Section 2.2), at least this variation of the protocol does not influence parameter choice. Moreover, in further studies, the method should be applied to diverse sports and activities. Another point of weakness is the number of subjects, which should be increased.

Further studies are also required for the AnT determination. Up to now, we did not find a clear criterion by using automatic RQA analyses for the anaerobic threshold AnT. In previous work [9], we defined a different approach with respect to the determinism minima criterion to assess by Ant-based. Further physiological observations and criteria, as those reported Section 2.3, could be considered to improve this analysis, which was performed only on heart rate time series recorded during a submaximal exercise. Possibly, a cluster analysis [25] could be employed as an additional step to classify and characterize *DET* minima.

## 5. Conclusions

The aim of this work was to detect in real time the first metabolic threshold, AerT, in a fully automated way under the hypothesis that AerT corresponds to the determinism minima characterized by the highest convexity. Despite the limits of this work, this method allows the detection of AerT, relying only on heart rate time series overcoming currently used visual inspection, and in the future, once validated for different activities, can be easily implemented in applications acquiring data from portable heart rate monitors.

## Figures and Tables

**Figure 1 ijerph-20-01998-f001:**
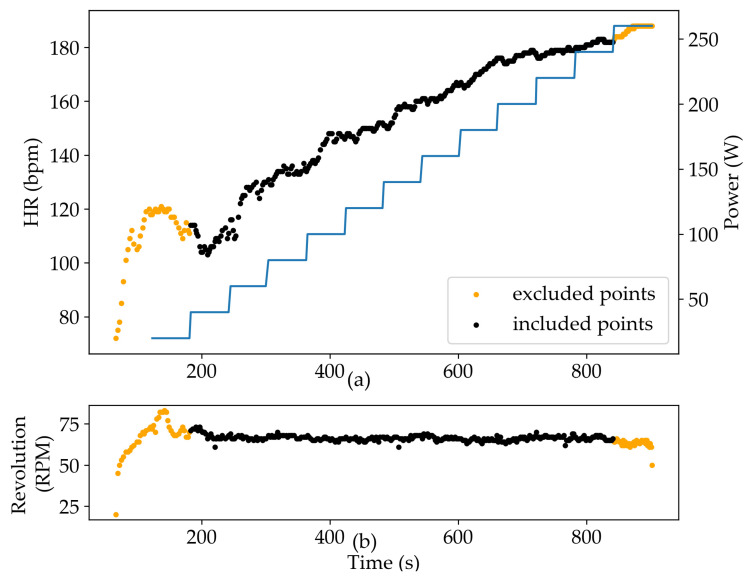
(**a**) Representative HR time series (bpm, black and orange dots) (subj #1) and the corresponding exercise intensity values (Watt, blue line). (**b**) Revolution time series (RPM, black and orange dots). Orange dots indicates workload increment intervals of the exercise that were excluded from the analysis.

**Figure 2 ijerph-20-01998-f002:**
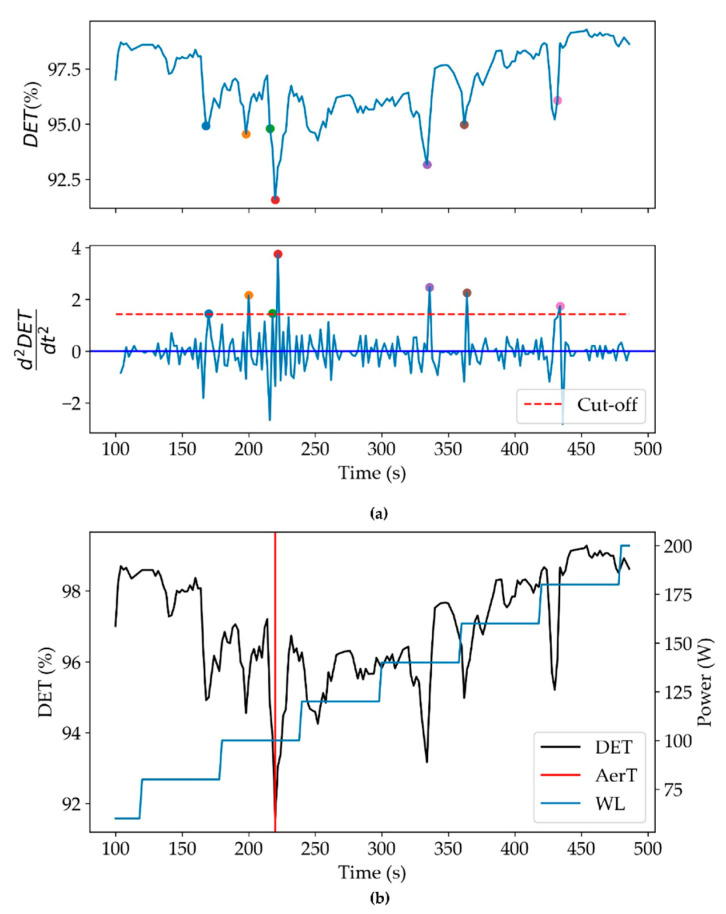
In a representative case (subj #2), (**a**) Percentage of determinism and $\frac{d^{2}DET}{dt^{2}}$; the dashed red horizontal line corresponds to the Cut-off; (**b**) percentage of determinism (DET, in black), and Workload (in blue) are shown point by point, respectively. The red vertical line corresponds to AerT (at time: 222 s).

**Figure 3 ijerph-20-01998-f003:**
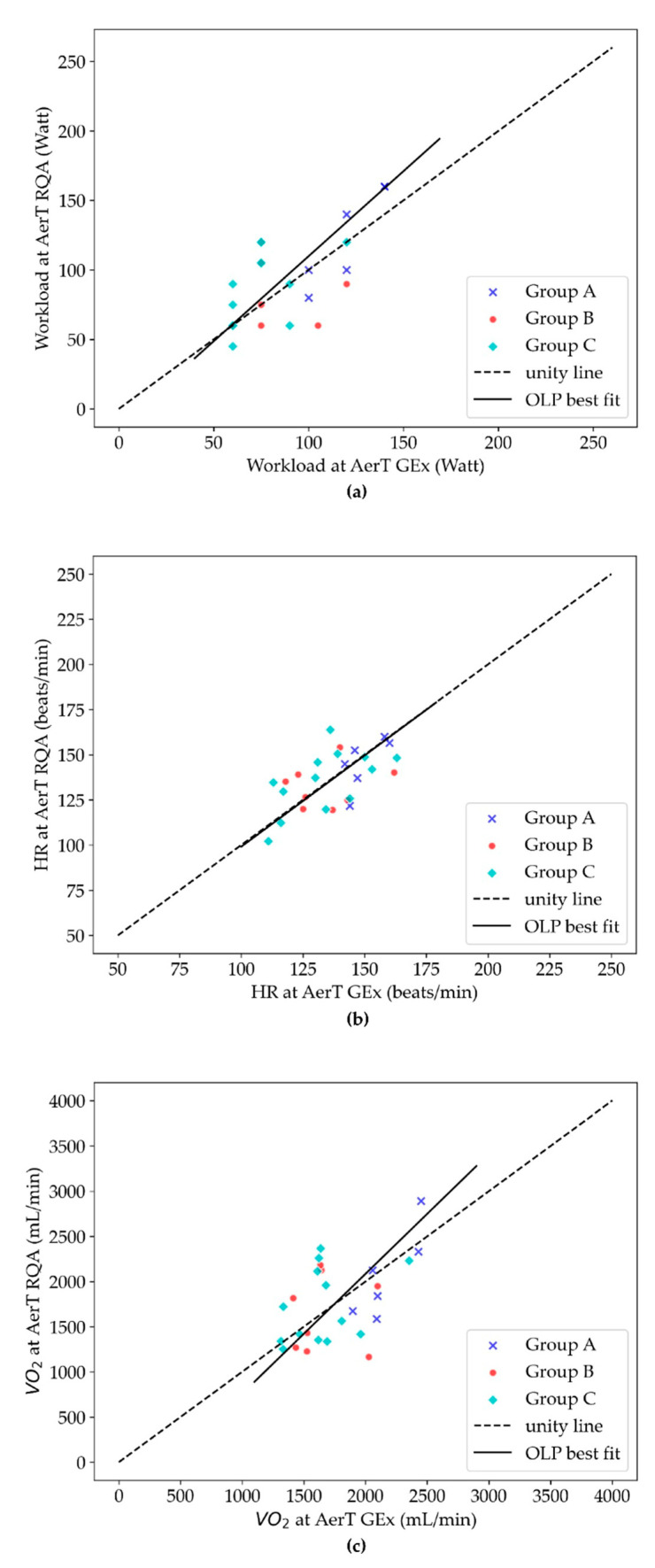
OLP regression line on (**a**) Workload y = 1.24 × −12.55; (**b**) HR y = 1.01 × −1.92; (**c**) VO_2_ y = 1.33 × −570.18 at AerT all values correspond to the mimima *DET*min. Group A (blue stars), B (red circles), and C (light blue diamonds).

**Table 1 ijerph-20-01998-t001:** Agreement with gold standard (RQA vs. GEx) for Workload (Watt), HR (bpm), and VO_2_ (mL/min) value.

Parameters	R^2^	Slope	Intercept	Mean Diff (%)	*p*	TE(%)
Workload (W)	0.41	1.22 (0.89 to 1.69)	−12.55 (−52.64 to 16.54)	10.27	0.16	14.52
HR (bpm)	0.32	1.01 (0.72 to 1.42)	−1.92 (−58.87 to 38.46)	0.11	0.85	7.51
VO_2_ (mL/min)	0.26	1.33 (0.93 to 1.9)	−570.18 (−1582.19 to 136.9)	1.70	0.91	16.25

All *p* > 0.05.

**Table 2 ijerph-20-01998-t002:** Agreement with gold standard (RQA vs. GEx) for Workload (Watt), HR (bpm), and VO_2_ (mL/min) value.

Parameters	r Pearson	ICC	Effect Size (d)
Workload (W)	0.64 **	0.77(0.50 to 0.90)	0.24
HR (bpm)	0.57 **	0.72(0.39 to 0.87)	0.04 *
VO_2_ (mL/min)	0.51 **	0.66(0.25 to 0.84)	0.02 *

* d < 0.2 trivial; (0.2 < d < 0.6 moderate), ** all large.

**Table 3 ijerph-20-01998-t003:** ANOVA 1-way on age and anthropometric measures.

Group	Height (cm)	Weight (kg)	Age (ys)
A (6)	180.13 ± 4.57	69.93 ± 3.76	15.33 ± 2.16
B (8)	174.63 ± 6.66	69.85 ± 11.73	16.00 ± 2.00
C (13)	173.72 ± 8.99	66.52 ± 16.63	14.77 ± 1.64
ABC (27)	175.41 ± 7.75	68.26 ± 13.05	15.26 ± 1.87
*p*	0.24 (F = 1.53)	0.81 (F = 0.21)	0.36 (F = 1.08)

All *p* > 0.05.

**Table 4 ijerph-20-01998-t004:** ANOVA 1-way on number of workload increments (Nstep), number of minima (Nmin), and number of points of HR time series (Npoints).

	Nstep	Nmin	Npoints
A (6)	8.83 ± 1.17	5.00 ± 2.07	93.75 ± 23.381
B (8)	9.63 ± 1.6	5.75 ± 2.19	81.92 ± 20.831
C (13)	8.21 ± 2.11	4.38 ± 2.36	93.15 ± 21.75
ABC (27)	8.81 ± 1.86	5.04 ± 2.26	116.67 ± 25.121
*p*	0.30 (F = 1.27)	0.359 (F = 1.07)	0.87 (F = 2.71)

All *p* > 0.05.

**Table 5 ijerph-20-01998-t005:** Workload, HR, and VO_2_ at AerT automatically detected by RQA-method in competitive rowers (A), recreational rowers (B), and athletes that practice other recreational sports (C).

Group	W_RQA (W)	W_GEx (W)	HR_RQA (bpm)	HR_RQA (bpm)	VO_2__RQA (mL/min)	VO_2__GEx (mL/min)
A (6)	123.33 ± 34.45	120.00 ± 17.89	145.42 ± 14.26	149.50 ± 7.58	2075.23 ± 487.85	2169.23 ± 220.90
B (8)	84.38 ± 23.97	82.50 ± 19.64	132.47 ± 11.95	134.25 ± 14.30	1646.54 ± 419.82	1662.35 ± 259.87
C (13)	85.38 ± 26.96	73.85 ± 17.81	135.49 ± 17.18	133.64 ± 16.31	1718.72 ± 411.59	1646.93 ± 284.44
ABC (27)	93.52 ± 31.34	86.67 ± 25.61	136.8 ± 15.41	137.34 ± 15.26	1776.56 ± 445.94	1767.57 ± 335.89
p method ^§^	0.29		0.66		0.80	
p method ^§^ xgroup	0.65		0.69		0.70	

**^§^** GEx method vs. RQA method; all *p* > 0.05, post-hoc power = 0.88.

## Data Availability

The data that support the findings of this study are available upon reasonable request from the authors.

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
