# Peer review of "Automatic Detection of Aerobic Threshold through Recurrence Quantification Analysis of Heart Rate Time Series"

_ijerph, 2023, doi:10.3390/ijerph20031998_

Round 1

Reviewer 1 Report

The topic is promising, the abstract section describes the problem, method and results, and the discussion section is clear. However, it is required to increase the resolution of all figures and add the conclusion section detailing the scope, limitations, contribution, applications and future works.

Author Response

Please download the document here enclosed

Reviewer 2 Report

Interesting study of methodological validation that proposes the automatic detection of the aerobic or ventilatory threshold from the analysis of the Determinism percentage, obtained through RQA in time series of heart rate, breath by breath. The study was carried out in a population sample of adolescents (12 to 20 years old), divided into three groups (n= 8,8,14), who underwent a cardiopulmonary exercise test on a bicycle ergometer.

Although the sample size per group was small, the statistical analysis of the data supports the findings that validate the automatic detection of the aerobic threshold in a manner comparable to the standard measurement through the inflection of the ventilatory equivalent (VE/VO2).

In general, the study is well planned in terms of the agreement among the objective with the methodology, results and conclusions. However, it is important to point out some deficiencies that must be considered with the intention of increasing the quality of the manuscript.

The main observation refers to the carelessness in the references provided, with one of them incomplete, others out of context and an excessive reference to the authors' own publications, even when these may not be the appropriate references for the topic being mentioned. An exhaustive review of each of the references is recommended, citing those that are a direct source of specific information on the topics covered in the manuscript text. It is also suggested to indicate an estimate of the sample size by group and specify the statistical power achieved.

SPECIFIC MINOR COMMENTS

- Could the results derived from the use of breath-by-breath heart rate be improved if beat-by-beat heart rate is used?

- Given the sample size, would it be preferable to use non-parametric statistical analysis?

- In the manuscript, the reason for not including data on the automatic detection of the anaerobic threshold is briefly justified. The argument given seems more like a limitation of the laboratory where the study was conducted than a physiological limitation. Seen in this way, it would be convenient to broaden the discussion or also include data on the anaerobic threshold.

- Table 1 (Appendix A) can be included within the text and should indicate the sample size (n) in each group.

- According to what is described in the Material and Methods section, is the total sample size 27 (see line 271) or 30 (8+8+14)?

- Figure B.1 (Appendix B) and Table C.1 (Appendix C) contain information already summarized in the text, so that in particular the data in Table C1 seem excessive. It is suggested to evaluate the possible elimination of both.

- Table D.1 (Appendix D) could also be included and commented on in the Materials and Methods section; In addition, it is worth discussing the importance of the number of points in the time series (Nstep, Nmin and Npoints).

- Line 44. Reference [1] is completely unrelated to the subject of the study under review; must be changed.

-Line 46. References [2] and [3] are not direct sources of the information mentioned; should be changed.

- Line 48. The citation in reference [4] is incomplete; must be corrected.

- Line 49. Reference [1] is completely unrelated to the subject of the study under review; must be changed.

- Line 53. Reference [2] is not a direct source of information on the topic being mentioned; must be changed.

- Line 56. Reference [3] may be acceptable, but reference [4] is incomplete; it is recommended to look for better references.

- Line 58. References [2] and [6] are not a direct source of information on the subject mentioned; better references must be sought.

- Line 59. Reference [7] is not a direct source of information on the topic being mentioned; a better reference should be found.

- Line 59. The citation in reference [4] is incomplete; must be corrected.

- Line 62. Reference [8] corresponds to the perception scale validation; search for a better reference on CPET guidelines.

- Lines 78-82. Statements in this introductory section describe in an anticipated and unnecessary way the main findings of the study, so it is suggested that they be eliminated or restated as a hypothesis.

- Lines 127-128. The criterion proposed in item 4) could be valid, but unnecessary. Reference [10], which is used as an argument for this criterion, indicates that the study was carried out on a treadmill and does not mention anything about a 20-minute criterion. It should be noted that the dynamics of the physiological response on a treadmill may be different from that of the exercise performed on a bicycle ergometer. It is recommended to eliminate or restate the wording of item 4) justifying the criterion with another reference.

- Line 128. Reference [10] corresponds to a formal publication in 2001. If it is decided to keep the same reference, the citation should be corrected as “Baldari C, Guidetti L. VO2max, ventilatory and anaerobic thresholds in rhythmic gymnasts and young female dancers. J Sports Med Phys Fitness. 2001 Jun;41(2):177-82. PMID: 11447359.”

- Line 144. References [6],[7],[11] and [12] are acceptable, but give the impression of an abuse of self-citation when there are references more directly related to the methodology for RQA or RQE.

- Line 184. Reference [17] corresponds to a study in which the OLP regression analysis is used, but it is not a direct source of information for the description of the tool. The reference must be changed.

- Line 374. It should say Table D.1 instead of C.1

Reviewer 3 Report

Formatting

-Standardize the title by either capitalizing 1st letter of each word or capitalizing just the 1st letter of 1st word.

-2nd affiliation spell out in full

-Suggest arranging keywords in alphabetical order

- The word “rover” does not sound accurate. Suggest replacing “Cyclist”

-Font sizes in figure labels were too large compared to font size in the text

-Minor grammatical errors were found throughout the paper. Please check the English grammar

e.g.

“participant undergo to”

“They then were required to”

 Abstract

-lack of related works, problem statement, and study objectives

 Introduction

-Line 59: “Despite their efficiency, the … laboratory equipped with specific devices and trained operators) [8]” Please state clearly what “their” is referring to.

-Line: 69:” Our group used RQA on heart rate time series to estimate the AerT in 69 obese subjects [7] and, successively, in healthy young subjects.” Do the authors refer to themselves as “our group”? Please replace “We” instead.

-The last paragraph – to move to the Methodology section.

 Method

Section 2.2

-Line 109- “They then were required to maintain a 60-70 revolutions per minute (rpm).” Please rephrase.

-Line 130- “Since …were not equal. HR was resampled.” The punctuation should be a comma “,”

Results

-Pls explain why R2 values are weak in Table 3, and R2 is stronger in Table 4. Does it mean the regressions are not well-fitted? Justify.

- Table 4 is not referred to in the text.

-Table 1 missing

-Table 2 is wrongly placed after Table 5

- Suggest graphs to include Legend for different line colours/bullet labels

 Discussion

-less convincing

-suggest providing evidence to support results in terms of significance/non-significant findings

 Conclusion?

-This section is not found

-Please include the Conclusion section
